# A Debiased MDI Feature Importance Measure for Random Forests

**Xiao Li**[*]
Statistics Department
UC Berkeley
sxli@berkeley.edu

**Yu Wang**
Statistics Department
UC Berkeley
wang.yu@berkeley.edu

**Sumanta Basu**
Statistics and Data Science Department
Computational Biology Department
Cornell University
sumbose@cornell.edu

**Karl Kumbier**
Statistics Department
UC Berkeley
kkumbier@berkeley.edu

**Bin Yu**
EECS, Statistics Department
UC Berkeley
binyu@berkeley.edu

## Abstract

Tree ensembles such as Random Forests have achieved impressive empirical success across a wide variety of applications. To understand how these models make predictions, people routinely turn to feature importance measures calculated from tree ensembles. It has long been known that Mean Decrease Impurity (MDI), one of the most widely used measures of feature importance, incorrectly assigns high importance to noisy features, leading to systematic bias in feature selection. In this paper, we address the feature selection bias of MDI from both theoretical and methodological perspectives. Based on the original definition of MDI by Breiman et al. [3] for a single tree, we derive a tight non-asymptotic bound on the expected bias of MDI importance of noisy features, showing that deep trees have higher (expected) feature selection bias than shallow ones. However, it is not clear how to reduce the bias of MDI using its existing analytical expression. We derive a new analytical expression for MDI, and based on this new expression, we are able to propose a new MDI feature importance measure using out-of-bag samples, called MDI-oob. For both the simulated data and a genomic ChIP dataset, MDI-oob achieves state-of-the-art performance in feature selection from Random Forests for both deep and shallow trees.

## 1 Introduction

Understanding how a machine learning (ML) model makes predictions is important in many scientific and industrial problems [19]. Appropriate interpretations can help increase the predictive performance of a model and provide new domain insights. While a line of study focuses on interpreting any generic ML model [30, 22], there is a growing interest in developing specialized methods to understand specific models. In particular, interpreting Random Forests (RFs) [2] and its variants [14, 28, 27, 29, 1, 12] has become an important area of research due to the wide ranging applications of RFs in various scientific areas, such as genome-wide association studies (GWAS) [7], gene expression microarray [13, 23], and gene regulatory networks [9].

A key question in understanding RFs is how to assign feature importance. That is, which features does a RF rely on for prediction? One of the most widely used feature importance measures for

---

[*]The first two authors contributed equally to this paper.

RFs is mean decrease impurity (MDI) [3]. MDI computes the total reduction in loss or impurity contributed by all splits for a given feature. This method is computationally very efficient and has been widely used in a variety of applications [25, 9]. However, theoretical analysis of MDI has remained sparse in the literature [11]. Assuming there are an infinite number of samples, Louppe et al. [16] characterized MDI for totally randomized trees using mutual information between features and the response. They showed that noisy features, i.e., features independent of the outcome, have zero MDI importance. However, empirical studies have shown that MDI systematically assigns higher feature importance values to numerical features or categorical features with many categories [29]. In other words, high MDI values do not always correspond to the predictive associations between features and the outcome. We call this phenomenon MDI *feature selection bias*. Louppe [15] studied this issue and demonstrate via simulations that early stopping mechanisms (e.g., limited depth and larger leaf sizes) are effective to reduce the feature selection bias.

In this paper, using the original definition of MDI, we analyze the non-asymptotic behavior of MDI and bridge the gap between the population case and the finite sample case. We find that under mild conditions, if the samples used for each tree are i.i.d, then the expected MDI feature importance of noisy features derived from any tree ensemble constructed on $n$ samples with $p$ features is upper bounded by $d_n \log(np)/m_n$, where $m_n$ is the minimum leaf size and $d_n$ is the maximum tree depth in the ensemble. In other words, deep trees with small leaves suffer more from feature selection bias. Our findings are particularly relevant for practical applications involving RFs, in which scenario deep trees are recommended [2] and used more often. To reduce the feature selection bias for RFs, especially when the trees are deep, we derive a new analytical expression for MDI and then use this new expression to propose a new feature importance measure that evaluates MDI using out-of-bag samples. We call this importance measure MDI-oob. For both regression and classification problems, we use simulated data and a genomic dataset to demonstrate that MDI-oob often achieves 5%–10% higher AUC scores compared to other feature importance measures used in several publicly available packages including `party` [4], `ranger` [33], and `scikit-learn` [21].

## 1.1 Related works

In addition to MDI [32, 17], some other feature importance measures have been studied in the literature and used in practice:

- Split count, namely, the number of times a feature is used to split [29], can be used as a feature importance measure. This method has been studied in [28, 1] and is available in XGBoost [6].
- Mean decrease in accuracy (MDA) measures a feature's importance by the reduction in the model's accuracy after randomly permuting the values of a feature. The motivation of MDA is that permuting an important feature would result in a large decrease in the accuracy while permuting an unimportant feature would have a negligible effect. Different permutation choices have been studied in [28, 10].

Recently, Lundberg et al. [17] show that for feature importance measures such as MDI and split counts, the importance of a feature does not always increase as the outcome becomes more dependent on that feature. To remedy this issue, they propose the tree SHAP feature importance, which focuses on giving consistent feature attributions to each sample. When individual feature importance is obtained, overall feature importance is straightforward to obtain by just averaging the individual feature importances across samples.

While our paper focuses on interpreting trees learned via the classic RF procedure, there is another line of work that focuses on modifying the tree construction procedure to obtain better feature importance measures. Hothorn et al. [8] introduced cforest in the R package `party` that grows classification trees based on a conditional inference framework. Strobl et al. [29] showed that cforest suffers less from the feature selection bias. Sandri and Zuccolotto [25] proposed to create a set of uninformative pseudo-covariates to evaluate the bias in Gini importance. Nembrini et al. [20] gave a modified algorithm that is faster than the original method proposed by Sandri and Zuccolotto [25] with almost no overhead over the creation of the original RFs and available in the R package `ranger`. In a very recent paper, Zhou and Hooker [34] proposed to evaluate the decrease in impurity at each node using out-of-bag samples. However, our implementation is different from that in [34] and MDI-oob enjoys higher computational efficiency.

In Section 4, we will compare MDI-oob with all the aforementioned methods except the split count, for which we did not find a package that implements it for RFs.

## 1.2 Organization

The rest of this paper is organized as follows. In Section 2, we provide a non-asymptotic analysis to quantify the bias in the MDI importance when noisy features are independent of relevant features. In Section 3, we give a new characterization of MDI and propose a new MDI feature importance using out-of-bag samples, which we call MDI-oob. In Section 4, we compare our MDI-oob with other commonly used feature importance measures in terms of feature selection accuracy using the simulated data and a genomic ChIP dataset. We conclude our work and discuss possible future directions in Section 5.

## 2 Understanding the feature selection bias of MDI

In this section, we focus on understanding the finite sample properties of MDI importance and why it may have a significant bias in feature selection. We first briefly review the construction of RFs and introduce some important notations. Then, using the original definition of MDI, we give a tight upper bound to quantify the expected bias of MDI importance for a noisy feature. This upper bound is tight up to a $\log n$ factor where $n$ is the number of i.i.d. samples.

### 2.1 Background and notations

Suppose that the data set $\mathcal{D}$ contains $n$ i.i.d samples from a random vector $(X_1, \ldots, X_p, Y)$, where $X = (X_1, \ldots, X_p) \in \mathbb{R}^p$ are $p$ input features and $Y \in \mathbb{R}$ is the response. The $i^{th}$ sample is denoted by $(\mathbf{x}_i, y_i)$, where $\mathbf{x}_i = (x_{i1}, \ldots, x_{ip})$. We say that a feature $X_k$ is a *noisy* feature if $X_k$ and $Y$ are independent, and a *relevant* feature otherwise. Note that this definition of noisy features has also been used in many previous papers such as [16, 26]. We denote $S \subset [p]$ as the set of indexes of relevant features. We are particularly interested in the case where the number of relevant features is small, namely, $|S|$ is much smaller than $p$. For any number $m \in \mathbb{N}$, $[m]$ denotes the set of integers $\{1, \ldots, m\}$. For any hyper-rectangle $R \subset \mathbb{R}^p$, let $\mathbb{1}(X \in R)$ be the indicator function taking value one when $X \in R$ and zero otherwise.

RFs are an ensemble of classification and regression trees, where each tree $T$ defines a mapping from the feature space to the response. Trees are constructed independently of one another on a bootstrapped or subsampled data set $\mathcal{D}^{(T)}$ of the original data $\mathcal{D}$. Any node $t$ in a tree $T$ represents a subset (usually a hyper-rectangle) $R_t$ of the feature space. A split of the node $t$ is a pair $(k, z)$ which divides the hyper-rectangle $R_t$ into two hyper-rectangles $R_t \cap \mathbb{1}(X_k \leq z)$ and $R_t \cap \mathbb{1}(X_k > z)$, corresponding to the left child $t^{\text{left}}$ and right child $t^{\text{right}}$ of node $t$, respectively. For a node $t$ in a tree $T$, $N_n(t) = |\{i \in \mathcal{D}^{(T)} : \mathbf{x}_i \in R_t\}|$ denotes the number of samples falling into $R_t$ and

$$\mu_n(t) := \frac{1}{N_n(t)} \sum_{i:\mathbf{x}_i \in R_t} y_i \tag{1}$$

denotes their average response.

Each tree $T$ is grown using a recursive procedure which proceeds in two steps for each node $t$. First, a subset $\mathcal{M} \subset [p]$ of features is chosen uniformly at random. Then the optimal split $v(t) \in \mathcal{M}, z(t) \in \mathbb{R}$ is determined by maximizing:

$$\Delta_{\mathcal{I}}(t) := \text{Impurity}(t) - \frac{N_n(t^{\text{left}})}{N_n(t)}\text{Impurity}(t^{\text{left}}) - \frac{N_n(t^{\text{right}})}{N_n(t)}\text{Impurity}(t^{\text{right}}) \tag{2}$$

for some impurity measure Impurity$(t)$. The procedure terminates at a node $t$ if two children contain too few samples, i.e., $\min\{N_n(t^{\text{left}}), N_n(t^{\text{right}})\} \leq m_n$ , or if all responses are identical. The threshold $m_n$ is called the *minimum leaf size*. If a node $t$ does not have any children, it is called a leaf node; otherwise, it is called an inner node. We define the set of inner nodes of a tree $T$ as $I(T)$. We say that $T'$ is a sub-tree of $T$ if $T'$ can be obtained by pruning some nodes in $T$.

Some popular choices of the impurity measure Impurity$(t)$ include variance, Gini index, or entropy. For simplicity, we focus on the variance of the responses, i.e.,

$$\text{Impurity}(t) = \frac{1}{N_n(t)} \sum_{i:\mathbf{x}_i \in R_t} (y_i - \mu_n(t))^2, \tag{3}$$

throughout the paper unless stated otherwise. Later we show that this definition of impurity is equivalent to the Gini index of categorical variables with one hot encoding (see Remark in Section 3)

The Mean Decrease Impurity (MDI) feature importance of $X_k$, with respect to a single tree $T$ (first proposed by Breiman et al. in [3]) and an ensemble of $n_{tree}$ trees $T_1, \ldots, T_{n_{tree}}$, can be written as

$$\text{MDI}(k,T) = \sum_{t \in I(T), v(t)=k} \frac{N_n(t)}{n} \Delta_\mathcal{I}(t) \quad \text{and} \quad \text{MDI}(k) = \frac{1}{n_{tree}} \sum_{s=1}^{n_{tree}} MDI(k, T_s), \quad (4)$$

respectively. This expression is the best known formula for MDI and was analyzed in many papers such as Louppe et al. [16].

## 2.2 Finite sample bias of MDI importance for Random Forests

Given the set $S$ of relevant features and a tree $T$, we denote

$$G_0(T) = \sum_{k \notin S} \text{MDI}(k, T) \quad (5)$$

as the sum of MDI importance of all noisy features. Ideally, $G_0(T)$ should be close to zero with high probability, to ensure that no noisy features get selected when using MDI importance for feature selection. In fact, Louppe et al. [16] show that $G_0(T)$ is indeed zero almost surely if we grow totally randomized trees with infinite samples. However, $G_0(T)$ is typically non-negligible in real data, and finite sample properties of $G_0(T)$ are not well understood. In order to bridge this gap, we conduct a non-asymptotic analysis of the expected value of $G_0(T)$. Our main result characterizes how the expected value of $G_0(T)$ depends on $m_n$, the minimum leaf size of $T$, and $p$, the dimension of the feature space. We start with the following simple but important fact.

**Fact 1.** *If $T'$ is a sub-tree of $T$, then $\text{MDI}(k, T') \leq \text{MDI}(k, T)$ for any feature $X_k$.*

This fact naturally follows from the observation that by definition, $\Delta_\mathcal{I}(t) \geq 0$ for any node $t$. Since the impurity decrease at each node is guaranteed to be non-negative, $G_0(T)$ will never decrease as $T$ grows deeper, in which case the minimum leaf size $m_n$ will be smaller. Indeed, if $T$ is grown to purity ($m_n = 1$), and all features are noisy ($S = \emptyset$), then $G_0(T)$ would simply be equal to the sample variance of the responses in the data $\mathcal{D}^{(T)}$. How fast does $G_0(T)$ increase as the minimum leaf size $m_n$ becomes smaller? To quantify the relation between $G_0(T)$ and $m_n$, we need a few mild conditions which we now describe. Let

$$y_i = \phi(\mathbf{x}_{i,S}) + \epsilon_i, i = 1, \ldots, n \quad (6)$$

for some unknown function $\phi : \mathbb{R}^{|S|} \to \mathbb{R}$, where $\epsilon_i$ are i.i.d zero-mean Gaussian noise. We make the following assumptions.

(A1) $X_k \sim \text{Unif}[0,1]$ for all $k \in [p]$. In addition, the noisy features $\{X_k, k \in [p]\backslash S\}$ are mutually independent, and independent of all relevant features. Here $S$ denotes the set of relevant features.

(A2) $\phi$ is bounded: $\sup_{\mathbf{x} \in [0,1]^{|S|}} |\phi(\mathbf{x})| \leq M$ for some $M > 0$.

The Assumptions (A1) and (A2) are weaker than the assumptions usually made when studying the statistical properties of RF. The marginal uniform distribution condition in (A1) is common in the RF literature [26], and can be easily satisfied by transforming the features via its inverse CDF. Since we are interested in characterizing the MDI of noisy features, we do not require the relevant features to be independent of each other. We do require that noisy features are independent of relevant features, which is a limitation of Theorem 1 below. Correlated features are commonly encountered in practice and difficult for any feature selection method.

We now state our first main result which provides a non-asymptotic upper and lower bound for the expected value of the maximum of $G_0(T)$ over all tree $T$ with minimum leaf size $m_n$.

**Theorem 1.** *Let $\mathcal{T}_n(m_n)$ denote the set of decision trees whose minimum leaf size is lower bounded by $m_n$, and $\mathcal{T}_n(m_n, d_n) \subset \mathcal{T}_n(m_n)$ denote the subset of $\mathcal{T}_n(m_n)$ whose depth is upper bounded by $d_n$. Under Assumptions (A1) and (A2), there exists a positive constant $C$ such that,*

$$\mathbb{E}_{X,\epsilon} \sup_{T \in \mathcal{T}_n(m_n, d_n)} G_0(T) \leq C \frac{d_n \log(np)}{m_n}. \quad (7)$$

*In addition, when $f = 0$ and $m_n \geq 36 \log p + 18 \log n$,*

$$\mathbb{E}_{X,\epsilon} \sup_{T \in \mathcal{T}_n(m_n)} G_0(T) \geq \frac{\log p}{Cm_n}. \tag{8}$$

We give the proof in the Appendix. To the best of our knowledge, Theorem 1 is the first non-asymptotic result on the expected MDI importance of tree ensembles. In particular, the upper bound can be directly applied to *any* tree ensembles with a minimum leaf size $m_n$ and a maximum tree depth $d_n$, including Breiman's original RF procedure, if subsampling is used instead of bootstrapping.

*Proof Sketch.* Every node $t$ in a tree $T \in \mathcal{T}_n(m_n, d_n)$ corresponds to an axis-aligned hyper-rectangle in $[0, 1]^p$ which contains at least $m_n$ samples and is formed by splitting on at most $d_n$ dimensions consecutively. Therefore, bounding the supremum of impurity reduction for any potential node in $\mathcal{T}_n(m_n, d_n)$ boils down to controlling the complexity of all such hyper-rectangles. Two hyper-rectangles are considered equivalent if they contain the same subset of samples, since the impurity reductions of these two hyper-rectangles are always the same. Up to this equivalence, it can be proved that the number of unique hyper-rectangles of interest is upper bounded by $(np)^{d_n}$, which corresponds to the $d_n log(np)$ term in the upper bound. The final result is obtained via union bound. □

In the upper bound, each node $t$ is obtained by splitting on at most $d_n$ features. In practice, $d_n$ is typically at most of order $\log n$. Indeed, if the decision tree is a balanced binary tree, then $d_n \leq \log_2 n$. Therefore, for balanced trees, the upper bound can be written as

$$\mathbb{E}_{X,\epsilon} \sup_{T \in \mathcal{T}_n(m_n, d_n)} G_0(T) \leq C \frac{d_n \log(np)}{m_n} \leq C \frac{(\log n)^2 + \log n \log p}{m_n}, \tag{9}$$

and the theorem shows that the sum of MDI importance of noisy features is of order $\frac{\log p}{m_n}$, i.e.,

$$\sup_{\phi : \|\phi\|_\infty \leq M} \mathbb{E}_{X,\epsilon} \sup_{T \in \mathcal{T}_n(m_n)} G_0(T) \sim \frac{\log p}{m_n}, \tag{10}$$

up to a $\log n$ term correction, which is typically small in the high dimensional $p \gg n$ setting. If all features $X_j$ are categorical with a bounded number of categories, then the upper bound can be improved to

$$\mathbb{E}_{X,\epsilon} \sup_{T \in \mathcal{T}_n(m_n, d_n)} G_0(T) \leq C \frac{d_n \log p}{m_n}, \tag{11}$$

which shows that the MDI importance of noisy features can be better controlled if the noisy features are categorical rather than numerical. That is consistent with the previous empirical studies because the number of candidate split points for a numerical feature is larger than that for a categorical feature.

Theorem 1 shows that the supremum of MDI importance of noisy features over all trees with minimum leaf size $m_n$ is, in expectation, roughly inversely proportional to $m_n$. In the Appendix Fig. 5, we show that the inversely proportional relationship is consistent with the empirical $G_0(T)$ on a simulated dataset described in the first simulation study in Section 4. Therefore, to control the finite sample bias of MDI importance, one should either grow shallow trees, or use only the shallow nodes in a deep tree when computing the feature importance. In fact, since $G_0(T)$ depends on the dimension $p$ only through a log factor $\log p$, we expect $G_0(T)$ to be very small even in a high-dimensional setting if $m_n$ is larger than, say, $\sqrt{n}$. For a balanced binary tree grown to purity with depth $d_n = \log_2 n$, this corresponds to computing MDI only from the first $d_n/2 = (\log_2 n)/2$ levels of the tree, as the node size on the $d$th level of a balanced tree is $n/2^d$.

Fact 1 implies that the MDI importance of relevant features might also decrease as $m_n$ increases, but we will show in simulation studies that they will decrease at a much slower rate, especially when the underlying model is sparse.

## 3 MDI using out-of-bag samples (MDI-oob)

As shown in the previous section, for balanced trees, the sum of MDI feature importance of all noisy features is of order $\frac{\log(p)}{m_n}$ if we ignore the $\log(n)$ terms. This means that the MDI feature selection bias becomes severe for trees with smaller leaf size $m_n$, which usually corresponds to a deeper tree.

Fortunately, this bias can be corrected by evaluating MDI using out-of-bag samples. In this section, we first introduce a new analytical expression of MDI as the motivation of our new method, then we propose the MDI-oob as a new feature importance measure. For simplicity, in this section, we only focus on one tree $T$. However, all the results are directly applicable to the forest case.

## 3.1 A new characterization of MDI

Recall that the original definition of the MDI importance of any feature $k$ is provided in Equation (4), that is, the sum of impurity decreases among all the inner nodes $t$ such that $v(t) = k$. Although we can use this definition to analyze the feature selection bias of MDI in Theorem 1, this expression (4) gives us few intuitions on how we can get a new feature importance measure that reduces the MDI bias. Next, we derive a novel analytical expression of MDI, which shows that the MDI of any feature $k$ can be viewed as the sample covariance between the response $y_i$ and the function $f_{T,k}(\mathbf{x}_i)$ defined in Proposition 1. This new expression inspires us to propose a new MDI feature importance measure by using the out-of-bag samples.

**Proposition 1.** *Define the function $f_{T,k}(\cdot)$ to be*

$$f_{T,k}(X) = \sum_{t \in I(T):v(t)=k} \left\{ \mu_n(t^{left})\mathbb{1}(X \in R_{t^{left}}) + \mu_n(t^{right})\mathbb{1}(X \in R_{t^{right}}) - \mu_n(t)\mathbb{1}(X \in R_t) \right\}.$$

*Then the MDI of the feature $k$ in a tree $T$ can be written as:*

$$\frac{1}{|\mathcal{D}^{(T)}|} \sum_{i \in \mathcal{D}^{(T)}} f_{T,k}(\mathbf{x}_i) \cdot y_i, \tag{12}$$

We give the proof in the Appendix. The proof is just a few lines but it requires a good understanding of MDI. Although we have not seen this analytical expression in the prior works, we found that the functions $f_{T,k}(\cdot)$ have been studied from a quite different perspective. Those functions were first proposed in Saabas [24] to interpret the RF predictions for each individual sample. According to this paper, $f_{T,k}$ can be viewed as the "contribution" made by the feature $k$ in the tree $T$. For any tree, those functions $f_{T,k}$ can be easily computed using the python package *treeinterpreter*.

It can be shown that $\sum_{i \in \mathcal{D}^{(T)}} f_{T,k}(\mathbf{x}_i) = 0$. That implies $\frac{1}{|\mathcal{D}^{(T)}|} \sum_{i \in \mathcal{D}^{(T)}} f_{T,k}(\mathbf{x}_i) \cdot y_i$ is essentially the sample covariance between $f_{T,k}(\mathbf{x}_i)$ and $y_i$ on the bootstrapped dataset $\mathcal{D}^{(T)}$. This indicates a potential drawback of MDI: RFs use the training data $\mathcal{D}^{(T)}$ to construct the functions $f_{T,k}(\cdot)$, then MDI uses the same data to evaluate the covariance between $y_i$ and $f_{T,k}(\mathbf{x}_i)$ in Equation (12).

**Remark:** So far we have only considered regression trees, and have defined the impurity at a node $t$ using the sample variance of responses. For classification trees, one may use Gini index as the measure of impurity. We point out that these two definitions of impurity are actually equivalent when we use a one-hot vector to represent the categorical response. Therefore, our results are directly applicable to the classification case. Suppose that $Y$ is a categorical variable which can take $D$ values $c_1, c_2, \ldots, c_D$. Let $p_d = \mathbb{P}(Y = c_d)$. Then the Gini index of $Y$ is $\text{Gini}(Y) = \sum_{d=1}^{D} p_d(1 - p_d)$. We define the one-hot encoding of $Y$ as a $D$-dimensional vector $\tilde{Y} = (\mathbb{1}(Y = c_1), \ldots, \mathbb{1}(Y = c_D))$. Then

$$\text{Var}(\tilde{Y}) = \|\tilde{Y} - \mathbb{E}\tilde{Y}\|_2^2 = \sum_{d=1}^{D}(\mathbb{E}\tilde{Y}_i^2 - (\mathbb{E}\tilde{Y}_i)^2) = \sum_{d=1}^{D}(\mathbb{E}\tilde{Y}_i - (\mathbb{E}\tilde{Y}_i)^2) = \sum_{d=1}^{D} p_d(1 - p_d) = \text{Gini}(Y),$$
$$\tag{13}$$

thereby showing that Gini index and variance are equivalent.

## 3.2 Evaluating MDI using out-of-bag samples

Proposition 1 suggests that we can calculate the covariance between $y_i$ and $f_{T,k}(\mathbf{x}_i)$ in Equation (12) using the out-of-bag samples $\mathcal{D} \backslash \mathcal{D}^{(T)}$:

$$\text{MDI-oob of feature } k = \frac{1}{|\mathcal{D} \backslash \mathcal{D}^{(T)}|} \sum_{i \in \mathcal{D} \backslash \mathcal{D}^{(T)}} f_{T,k}(\mathbf{x}_i) \cdot y_i. \tag{14}$$

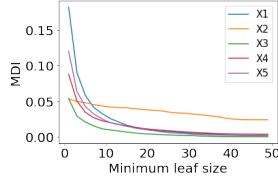 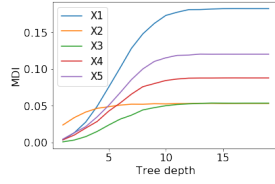 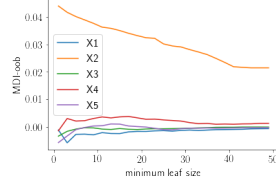

Figure 1: MDI against min leaf size.

Figure 2: MDI against tree depth.

Figure 3: MDI-oob against min leaf size.

In other words, for each tree, we calculate the $f_{T,k}(\mathbf{x}_i)$ for all the OOB samples $\mathbf{x}_i$ and then compute MDI-oob using (14). Although out-of-bag samples have been used for other feature importance measures such as MDA, to the best of the authors' knowledge, there are few results that use the out-of-bag samples to evaluate MDI feature importance. A naive way of using the out-of-bag samples to evaluate MDI is to directly compute the impurity decrease at each inner-node of a tree using OOB samples. However, this approach is not desirable since the impurity decrease at each node is still always positive unless the responses of all the OOB samples falling into a node are constant. In this case, an argument similar to the proof of Theorem 1 can show that the bias of directly computing impurity using OOB samples could still be large for deep trees. The idea of MDI-oob depends heavily on the new analytical MDI expression. Without the new expression, it is not clear how one can use out-of-bag samples to get a better estimate of MDI. One highlight of the MDI-oob is its low computation cost. The time complexity of evaluating MDI-oob for RFs is roughly the same as computing the RF predictions for $|\mathcal{D}\backslash\mathcal{D}^{(T)}|$ number of samples.

## 4 Simulation experiments[2]

**Simulated study on the effect of minimum leaf size and the tree depth**

In this simulation, we investigate the empirical relationship between MDI importance and the minimum leaf size. To mimic the major experiment setting in the paper [29], we generate the data as follows. We sample $n = 200$ observations, each containing 5 features. The first feature is generated from standard Gaussian distribution. The second feature is generated from a Bernoulli distribution with $p = 0.5$. The third/fourth/fifth features have 4/10/20 categories respectively with equal probability of taking any states. The response label y is generated from a Bernoulli distribution such that $P(y_i = 1) = (1 + x_{i2})/3$. While keeping the number of trees to be 300, we vary the minimum leaf size of RF from 1 to 50 and record the MDI of every feature. The results are shown in Fig. 1. We can see from this figure that the MDI of noisy features, namely X1, X3, X4 and X5, drops significantly when the minimum leaf size increases from 1 to 50. This observation supports our theoretical result in Theorem 1. Besides the minimum leaf size, we also investigate the relationship between MDI and the tree depth. As tree depth increases, the minimum leaf size generally decreases exponentially. Therefore, we expect the MDI of noisy features to become larger for increasing tree depth. We vary the maximum depth from 1 to 20 and record the MDI of every feature. The results shown in Fig. 2 are consistent with our expectation. MDI importance of noisy features increase when the tree depth increases from 1 to 20. Fig. 3 shows the MDI-oob measure and it indeed reduces the bias of MDI in this simulation.

**Noisy feature identification using the simulated data**

In this experiment, we evaluate different feature importance measures in terms of their abilities to identify noisy features in a simulated data set. We compare our method with the following methods: MDA, cforest in the R package `party`, SHAP[17], default feature importance (MDI) in `scikit-learn`, the impurity corrected Gini importance in the R package `ranger`, UFI in [34], and naive-oob, which refers to the naive method that evaluates impurity decrease at each node using out-of-bag samples directly. To evaluate feature importance measures, we generate the following simulated data. Inspired by the experiment settings in Strobl et al. [29], our setting involves discrete features with different number of distinct values, which poses a critical challenge for MDI. The data has 1000 samples with 50 features. All features are discrete, with the $j^{th}$ feature containing $j + 1$

distinct values $0, 1, \ldots, j$. We randomly select a set $S$ of 5 features from the first ten as relevant features. The remaining features are noisy features. Choosing active features with fewer categories represents the most challenging case for MDI. All samples are i.i.d. and all features are independent. We generate the outcomes using the following rules:

- Classification: $P(Y = 1|X) = \text{Logistic}(\frac{2}{5} \sum_{j \in S} X_j/j - 1)$.
- Regression: $Y = \frac{1}{5} \sum_{j \in S} X_j/j + \epsilon$, where $\epsilon \sim \mathcal{N}(0, 100 \cdot \text{Var}(\frac{1}{5} \sum_{j \in S} X_j/j))$.

Treating the noisy features as label 0 and the relevant features as label 1, we can evaluate a feature importance measure in terms of its area under the receiver operating characteristic curve (AUC). Note that when a feature importance measure gives low importance to relevant features, its AUC score measure can be smaller than 0.5 or even 0. We grow 100 trees with the minimum leaf size set to either 100 (shallow tree case) or 1 (deep tree case). The number of candidate features $m_{try}$ is set to be 10. We repeat the whole process 40 times and report the average AUC scores for each method in Table 1. The boxplots For this simulated setting, MDI-oob achieves the best AUC score under all cases.

**Noisy feature identification using a genomic ChIP dataset**

To evaluate our method MDI-oob in a more realistic setting, we consider a ChIP-chip and ChIP-seq dataset measuring the enrichment of 80 biomolecules at 3912 regions of the *Drosophila* genome [5, 18]. These data have previously been used in conjunction with RF-based methods, namely iterative random forests (iRF) [1], to predict functional labels associated with genomic regions. They provide a realistic representation of many issues encountered in practice, such as heterogeneity and dependencies among features, which make it especially challenging for feature selection problems. To evaluate feature selection in the ChIP data, we scale each feature $X_j$ to be between 0 and 1. Second, we randomly select a set $S$ of 5 features as relevant features and include the rest as noisy features. We randomly permute values of any noisy features to break their dependencies with relevant features. By this means, we avoid the cases where RFs "think" some features are important not because they themselves are important but because they are highly correlated with other relevant features. Then we generate responses using the following rules:

- Classification: $P(Y = 1|X) = \text{Logistic}(\frac{2}{5} \sum_{j \in S} X_j - 1)$.
- Regression: $Y = \frac{1}{5} \sum_{j \in S} X_j + \epsilon$, where $\epsilon \sim \mathcal{N}(0, 100 \cdot \text{Var}(\frac{1}{5} \sum_{j \in S} X_j))$.

All the other settings remain the same as the previous simulations. We report the average AUC scores for each method in Table 1. The standard errors and the beeswarm plots of all the methods are included in the Appendix. Naive-oob, namely, the method that directly computes MDI using the out-of-bag samples is hardly any better than the original gini importance. MDI-oob or UFI usually achieves the best AUC score in three out of four cases, except for shallow regression trees, when all methods appear to be equally good with AUC scores close to 1. Although UFI and MDI-oob use out-of-bag samples in different ways, their results are generally comparable. We also note that increasing the minimum leaf size consistently improves the AUC scores of all methods.

Another observation is that MDA behaves poorly in some simulations despite its use of a validation set. This could be due to the low signal-to-noise ratio in the simulation setting. After we train the RF model on the training set, we evaluated the model's accuracy on a test set. It turns out that the accuracy of the model is quite low. In that case, MDA struggles because the accuracy difference between permuting a relevant feature and permuting a noisy feature is small. We observe that the MDA gets better when we increase the signal-to-noise ratio.

The computation time of different methods is hard to compare due to a few factors. Because the packages including `scikit-learn` and `ranger` compute feature importance when constructing the tree, it is hard to disentangle the time taken to construct the trees and the time taken to get the feature importance. Furthermore, different packages are implemented in different programming languages so it is not clear if the time difference is because of the algorithm or because of the language. We implement MDI-oob in Python and for our first simulated classification setting, MDI-oob takes $\sim 3.8$ seconds for each run. To compare, `scikit-learn` which uses Cython (A C extension for Python) takes $\sim 1.4$ seconds to construct the RFs for each run. Thus, MDI-oob runs in a reasonable time frame and we expect it to be faster if it is implemented in C or C++.

Table 1: Average AUC scores for noisy feature identification

| | Deep tree (min leaf size = 1) | | | | Shallow tree(min leaf size = 100) | | | |
| | Simulated | | ChIP | | Simulated | | ChIP | |
| | C | R | C | R | C | R | C | R |
| --- | --- | --- | --- | --- | --- | --- | --- | --- |
| MDI-oob | **0.76** | 0.52 | 0.87 | 0.98 | **0.75** | **0.58** | **0.94** | 0.98 |
| UFI | 0.72 | **0.54** | **0.88** | **0.99** | **0.75** | 0.56 | **0.94** | 0.98 |
| naive-oob | 0.18 | 0.10 | 0.67 | 0.71 | 0.60 | 0.39 | 0.89 | 0.97 |
| SHAP | 0.55 | 0.33 | 0.82 | 0.96 | 0.68 | 0.46 | 0.91 | 0.97 |
| ranger | 0.56 | 0.50 | 0.73 | 0.97 | 0.55 | 0.49 | 0.76 | **0.99** |
| MDA | 0.49 | 0.51 | 0.54 | 0.97 | 0.50 | **0.58** | 0.50 | **0.99** |
| cforest | 0.65 | 0.50 | 0.79 | 0.93 | 0.70 | 0.49 | 0.90 | 0.98 |
| MDI | 0.12 | 0.09 | 0.60 | 0.71 | 0.63 | 0.40 | 0.88 | 0.97 |

"C" stands for classification, "R" stands for regression. The column maximum is bolded.

## 5    Discussion and future directions

Mean Decrease Impurity (MDI) is widely used to assess feature importance and its bias in feature selection is well known. Based on the original definition of MDI, we show that its expected bias is upper bounded by an expression that is inversely proportional to the minimum leaf size under mild conditions, which means deep trees generally have a higher feature selection bias than shallow trees. To reduce the bias, we derive a new analytical expression for MDI and use the new expression to obtain MDI-oob. For the simulated data and a genomic ChIP dataset, MDI-oob has exhibited the state-of-the-art feature selection performance in terms of AUC scores.

*Comparison to SHAP.* SHAP originates from game theory and offers a novel perspective to analyze the existing methods. While it is desirable to have 'consistency, missingness and local accuracy', our analysis indicates that there are other theoretical properties that are also worth taking into account. As shown in our simulation, the feature selection bias of SHAP increases with the depth of the tree, and we believe SHAP can also use OOB samples to improve feature selection performance.

*Relationship to honest estimation.* Honest estimation is an important technique built on the core notion of sample splitting. It has been successfully used in causal inference and other areas to mitigate the concern of over-fitting in complex learners due to usage of same data in different stages of training. The proposed algorithm MDI-oob has important connections with "honest sampling" or "honest estimation". For example, in Breiman's 1984 book [3], he proposed to use a separate validation set for pruning and uncertainty estimation. In [31], each within-leaf prediction is estimated using a different sub-sample (such as OOB sample) than the one used to decide split points. Theoretical results of these papers and Proposition 1 of our paper convey the same message, that finite sample bias is caused by using the same data for growing trees and for estimation, and the bias can be reduced if we leverage OOB data. We believe the theoretical contributions of those papers can also help us analyze the statistical properties (such as variance) of the MDI-oob.

*Future directions.* Although the MDI-oob shows promising results for selecting relevant features, it also raises many interesting questions to be considered in the future. First of all, how can MDI-oob be extended to better accommodate correlated features? Going beyond feature selection, can importance measures also rank the relevant features in a reasonable order? Finally, can we use the new analytical expression of MDI to give a tighter theoretical bound for MDI's feature selection bias? We are exploring these interesting questions in our ongoing work.

## Acknowledgements

The authors would like to thank Merle Behr and Raaz Dwivedi from University of California, Berkeley for their very helpful comments of this paper that greatly improve its presentation. Partial supports are gratefully acknowledged from ARO grant W911NF1710005, ONR grant N00014-16-1-2664, NSF grants DMS-1613002 and IIS 1741340, and the Center for Science of Information (CSoI), a US NSF Science and Technology Center, under grant agreement CCF-0939370.

## Footnotes

[2]The source code is available at `https://github.com/shifwang/paper-debiased-feature-importance`

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
