[Supplementary Material]

# Appendix: Proofs

*Proof of Theorem 1.* To state the proof of the theorem, we need to define more notations. For a generic set $A \subset [0,1]^p$, with slight abuse of notations, let $N_n(A) = \sum_i \mathbb{1}(\mathbf{x}_i \in A)$ be the number of samples with input features in $A$, and

$$\mu_n(A) = \frac{\sum_{\mathbf{x}_i \in A} y_i}{N_n(A)}$$

be the average response of those samples. For any feature $X_k$ and $z \in (0,1)$, let $\Delta_{\mathcal{I}}(A, (k,z))$ be the impurity decrease when splitting $A$ into $A \cap \{X_k \leq z\}$ and $A \cap \{z < X_k\}$, and $\Delta_{\mathcal{I}}(A, k) = \sup_{0 \leq z \leq 1} \Delta_{\mathcal{I}}(A, (k,z))$.

The proof of the theorem proceeds in three parts. First, we prove a lemma which gives a tail bound for $\Delta_{\mathcal{I}}(A, k)$. Second, we use the lemma and union bound to derive the upper bound for the expectation of $G_0(T)$. Finally, we use a separate argument based on Gaussian comparison inequalities to obtain the lower bound.

**Lemma 1.** *For any axis-aligned hyper-rectangle $A \subset [0,1]^p$, $k \notin S$ and $\delta > 0$, we have*

$$\mathbb{P}_{X,\epsilon}(\Delta_{\mathcal{I}}(A, k) \geq \delta | N_n(A)) \leq 4 N_n(A) e^{-\frac{\delta N_n(A)}{4(M+1)^2}}.$$

*Proof of Lemma 1.* We suppose without loss of generality that $\mathbf{x}_1, \ldots, \mathbf{x}_{N_n(A)} \in A$. For any $z \in [0,1]$, we let

$$A^{\text{left}} = A \cap \{0 \leq X_k \leq z\}, \quad A^{\text{right}} = A \cap \{z < X_k \leq 1\},$$

and introduce the shorthands

$$p^{\text{left}} = \frac{N_n(A^{\text{left}})}{N_n(A)}, \quad p^{\text{right}} = \frac{N_n(A^{\text{right}})}{N_n(A)}, \quad \mu^{\text{left}} = \mu_n(A^{\text{left}}), \quad \mu^{\text{right}} = \mu_n(A^{\text{right}}).$$

Then

$$\begin{aligned}
\Delta_{\mathcal{I}}(A, (k,z)) &= \frac{1}{N_n(A)} \sum_{\mathbf{x}_i \in A} (y_i - \mu_n(A))^2 - \frac{1}{N_n(A)} \sum_{\mathbf{x}_i \in A} (y_i - \mu_n(A^{\text{left}}))^2 \mathbb{1}(x_{ik} \leq z) \\
&\quad - \frac{1}{N_n(A)} \sum_{\mathbf{x}_i \in A} (y_i - \mu_n(A^{\text{right}}))^2 \mathbb{1}(x_{ik} > z) \\
&= \frac{1}{N_n(A)} \sum_{\mathbf{x}_i \in A} y_i^2 - \mu_n(A)^2 - p^{\text{left}}\left(\frac{1}{N_n(A)p^{\text{left}}} \sum_{\mathbf{x}_i \in A} y_i^2 \mathbb{1}(x_{ik} \leq z) - (\mu^{\text{left}})^2\right) \\
&\quad - p^{\text{right}}\left(\frac{1}{N_n(A)p^{\text{right}}} \sum_{\mathbf{x}_i \in A} y_i^2 \mathbb{1}(x_{ik} > z) - (\mu^{\text{right}})^2\right) \\
&= p^{\text{left}}(\mu^{\text{left}})^2 + p^{\text{right}}(\mu^{\text{right}})^2 - \mu_n(A)^2 \\
&= (p^{\text{left}}(\mu^{\text{left}})^2 + p^{\text{right}}(\mu^{\text{right}})^2)(p^{\text{left}} + p^{\text{right}}) - (p^{\text{left}}\mu^{\text{left}} + p^{\text{right}}\mu^{\text{right}})^2 \\
&= p^{\text{left}}p^{\text{right}}(\mu^{\text{left}} - \mu^{\text{right}})^2 \\
&\leq 2p^{\text{left}}p^{\text{right}}[(\mu^{\text{left}} - \mu)^2 + (\mu^{\text{right}} - \mu)^2] \\
&\leq 2p^{\text{left}}(\mu^{\text{left}} - \mu)^2 + 2p^{\text{right}}(\mu^{\text{right}} - \mu)^2,
\end{aligned}$$

where

$$\mu = \mathbb{E}[Y | X \in A] = \mathbb{E}[\phi(X) | X \in A].$$

Now suppose without loss of generality that $x_{1k} < x_{2k} < \cdots < x_{nk}$ (otherwise we can reorder the samples by $X_k$). Since $k \notin S$, $X_k$ is independent of $X_S$ and therefore independent of $Y$. Thus the distribution of $(y_1, \ldots, y_n)$ does not change after the reordering, i.e.,

$$y_i \overset{i.i.d}{\sim} (\phi(X) | X \in A) + \epsilon.$$

Note that

$$\sup_z p^{\text{left}}(\mu^{\text{left}} - \mu)^2 \leq \sup_{1 \leq m \leq N_n(A)} \frac{m}{N_n(A)} \left(\frac{1}{m} \sum_{i=1}^m y_i - \mu\right)^2.$$

Note that $Y$ is sub-Gaussian with parameter $M+1$. Therefore, for each $1 \leq m \leq N_n(A)$, by Hoeffding bound,

$$\mathbb{P}\left(\frac{m}{N_n(A)}\left(\frac{1}{m}\sum_{i=1}^{m} y_i - \mu\right)^2 \geq \delta \Big| N_n(A)\right) \leq 2e^{-(M+1)^2 \delta N_n(A)^2/m} \leq 2e^{-\frac{\delta N_n(A)}{(M+1)^2}}.$$

Therefore

$$\mathbb{P}\left(\sup_z p^{\text{left}}(\mu^{\text{left}} - \mu)^2 \geq \delta \Big| N_n(A)\right) \leq 2N_n(A)e^{-\frac{\delta N_n(A)}{(M+1)^2}}.$$

By symmetry, the same bound holds for $p^{\text{right}}(\mu^{\text{right}} - \mu)^2$. Therefore

$$\mathbb{P}(\Delta_\mathcal{I}(A, k) \geq \delta | N_n(A))$$
$$\leq \mathbb{P}\left(\sup_z p^{\text{left}}(\mu^{\text{left}} - \mu)^2 \geq \delta/2 | N_n(A)\right) + \mathbb{P}\left(\sup_z p^{\text{right}}(\mu^{\text{right}} - \mu)^2 \geq \delta/2 | N_n(A)\right)$$
$$\leq 4N_n(A)e^{-\frac{\delta N_n(A)}{4(M+1)^2}},$$

and the lemma is proved. $\qquad\qquad\square$

**Proof of the upper bound in Theorem 1**

Without loss of generality, assume that when we split on feature $k$, the cut is always performed along the direction of $k$ at some data point (and that data point falls into the right sub-tree). Suppose that $\epsilon_i$ has unit variance for all $i$. Let $C = 2\max\{256, 16(M+1)^2\}$. We also assume, without loss of generality, that $m_n \geq 8d_n$. Otherwise, since $G_0(T)$ is, by definition, upper bounded by the sample variance of $y$, we have

$$\mathbb{E}_{X,\epsilon}\sup_{T \in \mathcal{T}_n(m_n, d_n)} G_0(T) \leq \text{Var}(Y) \leq M^2 + 1 \leq 16(M+1)^2 \frac{d_n \log np}{m_n}.$$

To simplify notation, we define $\mathbf{x}_{n+1} = (0, \ldots, 0)$ and $\mathbf{x}_{n+2} = (1, \ldots, 1)$. For any $V \subset [p], \mathcal{L}, \mathcal{R} \in [n+2]^{|V|}$, let

$$A(V, \mathcal{L}, \mathcal{R}) = \{X = (X_1, \ldots, X_p) : x_{\mathcal{L}_i, V_i} \leq X_{V_i} < x_{\mathcal{R}_i, V_i}, 1 \leq i \leq |V|, 0 \leq X_k \leq 1, k \notin V\}$$

be the random axis-aligned hyper-rectangle obtained by splitting on features in $V$, where the left and right endpoints of the $i$th feature $V_i$ are determined by $x_{\mathcal{L}_i, V_i}$ and $x_{\mathcal{R}_i, V_i}$. Note that in this definition, we treat $\mathbf{x}_i$ as random variables rather than fixed, and $A(V, \mathcal{L}, \mathcal{R})$ can be the empty set with non-zero probability. Let

$$A(V) = \{A(V, \mathcal{L}, \mathcal{R}) | \mathcal{L}, \mathcal{R} \in [n+2]^{|V|}\}$$

be all axis-aligned hyper-rectangles obtained by splitting on features in $V$. For any $d \leq d_n$, let

$$A_d = \cup_{|V|=d} A(V)$$

be the collection of all possible subsets of $[0, 1]^p$ obtained by splitting on $d$ features.

Fix $\delta > \frac{96M^2 d_n}{m_n}$. We will first show that

$$\mathbb{P}_{X,\epsilon}\left(\exists A \in A_d, k \notin S : \Delta_\mathcal{I}(A, k) \geq \frac{m_n \delta}{N_n(A)} \text{ and } N_n(A) \geq m_n\right)$$
$$\leq 5(np)^{d+1} \exp\left(-\frac{\delta m_n}{\max\{256, 16(M+1)^2\}}\right). \tag{15}$$

Note that for any two events $C_1$ and $C_2$, the inequality $\mathbb{P}(C_1 \cap C_2) \leq \mathbb{P}(C_1 | C_2)$ always holds. Therefore, for any hyper-rectangle $A$, we have

$$\mathbb{P}_{X,\epsilon}\left(\Delta_\mathcal{I}(A, k) \geq \frac{m_n \delta}{N_n(A)} \text{ and } N_n(A) \geq m_n\right)$$
$$\leq \mathbb{P}_{X,\epsilon}\left(\Delta_\mathcal{I}(A, k) \geq \frac{m_n \delta}{N_n(A)} \Big| N_n(A) \geq m_n\right) \tag{16}$$

To simplify notation, we will drop the conditional event $N_n(A) \geq m_n$ in the remainder of the proof of the upper bound, unless stated otherwise.

Fix $V \subset [p], \mathcal{L}, \mathcal{R} \in [n+2]^{|V|}$, and $k \notin S$. Conditional on samples in $\mathcal{L}$ and $\mathcal{R}$, we would like to apply Lemma 1 to $A(V, \mathcal{L}, \mathcal{R})$ and $k$. The only problem is that there are now samples on the boundary of $A(V, \mathcal{L}, \mathcal{R})$, namely those in $\mathcal{L}$ and $\mathcal{R}$. Let $\mathbf{x}_{\mathcal{L}} = \{\mathbf{x}_i\}_{i \in \mathcal{L}}$ and $\mathbf{x}_{\mathcal{R}} = \{\mathbf{x}_i\}_{i \in \mathcal{R}}$. Conditional on $\mathbf{x}_{\mathcal{L}}$, $\mathbf{x}_{\mathcal{R}}$ and $N_n(A(V, \mathcal{L}, \mathcal{R}))$, and on the random variable $X \in A(V, \mathcal{L}, \mathcal{R})$, $X$ is uniformly distributed in $A(V, \mathcal{L}, \mathcal{R})$. For a set $A$, we let $A^\circ$ be the interior of $A$ and let $\bar{A}$ be the boundary of $A$. Since $m_n \geq 8d_n$,

$$\frac{N_n(A^\circ(V, \mathcal{L}, \mathcal{R}))}{N_n(A(V, \mathcal{L}, \mathcal{R}))} \geq \frac{m_n - 2d_n}{m_n} \geq \frac{3}{4}.$$

By Lemma 1, we have

$$\mathbb{P}_{X,\epsilon}\left(\Delta_{\mathcal{I}}(A^\circ(V, \mathcal{L}, \mathcal{R}), k) \geq \frac{m_n \delta}{3 N_n(A(V, \mathcal{L}, \mathcal{R}))} \Big| \mathbf{x}_{\mathcal{L}}, \mathbf{x}_{\mathcal{R}}, N_n(A(V, \mathcal{L}, \mathcal{R}))\right)$$
$$\leq 4 N_n(A^\circ(V, \mathcal{L}, \mathcal{R})) \exp\left(-\frac{\delta m_n N_n(A^\circ(V, \mathcal{L}, \mathcal{R}))}{12(M+1)^2 N_n(A(V, \mathcal{L}, \mathcal{R}))}\right) \qquad (17)$$
$$\leq 4n \exp\left(-\frac{\delta m_n}{16(M+1)^2}\right)$$

for large $n$. Since the right hand side does not depend on $\mathbf{x}_{\mathcal{L}}, \mathbf{x}_{\mathcal{R}}, N_n(A(V, \mathcal{L}, \mathcal{R}))$, we can take expectation with respect to them, and obtain

$$\mathbb{P}_{X,\epsilon}\left(\Delta_{\mathcal{I}}(A^\circ(V, \mathcal{L}, \mathcal{R}), k) \geq \frac{m_n \delta}{3 N_n(A(V, \mathcal{L}, \mathcal{R}))}\right) \leq 4n \exp\left(-\frac{\delta m_n}{16(M+1)^2}\right) \qquad (18)$$

On the other hand, we have the inequality

$$\Delta_{\mathcal{I}}(A(V, \mathcal{L}, \mathcal{R}), k) \leq \Delta_{\mathcal{I}}(A^\circ(V, \mathcal{L}, \mathcal{R}), k) + \frac{\sum_{i \in \mathcal{L}, \mathcal{R}} (y_i - \mu_n(A(V, \mathcal{L}, \mathcal{R})))^2}{N_n(A(V, \mathcal{L}, \mathcal{R}))}$$
$$\leq \Delta_{\mathcal{I}}(A^\circ(V, \mathcal{L}, \mathcal{R}), k) + \frac{\sum_{i \in \mathcal{L}, \mathcal{R}} 2(y_i^2 + \mu_n(A(V, \mathcal{L}, \mathcal{R}))^2)}{N_n(A(V, \mathcal{L}, \mathcal{R}))}. \qquad (19)$$

We have

$$\mathbb{P}_{X,\epsilon}\left(\frac{\sum_{i \in \mathcal{L}, \mathcal{R}} 2y_i^2}{N_n(A(V, \mathcal{L}, \mathcal{R}))} \geq \frac{m_n \delta}{3 N_n(A(V, \mathcal{L}, \mathcal{R}))}\right)$$
$$\leq \mathbb{P}\left(\frac{\sum_{i \in \mathcal{L}, \mathcal{R}} 4(f^2(\mathbf{x}_i) + \epsilon_i^2)}{N_n(A(V, \mathcal{L}, \mathcal{R}))} \geq \frac{m_n \delta}{3 N_n(A(V, \mathcal{L}, \mathcal{R}))}\right)$$
$$\leq \mathbb{P}\left(\frac{\sum_{i \in \mathcal{L}, \mathcal{R}} 4(f^2(\mathbf{x}_i) + \epsilon_i^2)}{m_n} \geq \frac{\delta}{3}\right) \qquad (20)$$
$$\leq \mathbb{P}\left(\frac{\sum_{i \in \mathcal{L}, \mathcal{R}} 4M^2 + 4\epsilon_i^2}{m_n} \geq \frac{\delta}{3}\right)$$
$$\leq \mathbb{P}\left(\frac{\sum_{i=1}^{2d_n}(\epsilon_i^2 - 1)}{m_n} \geq \frac{\delta}{16 N_n(A^\circ(V, \mathcal{L}, \mathcal{R}))}\right) \leq \exp(-\frac{\delta m_n}{256}),$$

for large $n$, where the fourth inequality holds because $\delta \geq 96 M^2 d_n / m_n$, and the last inequality follows from the well-known tail bound

$$\mathbb{P}\left(\left|\frac{1}{d}\chi_d^2 - 1\right| \geq \delta_0\right) \leq 2e^{-d\delta_0^2/8}$$

for $\chi_d^2$ random variable and $\delta_0 < 1$. To upper bound $\mu_n(A(V, \mathcal{L}, \mathcal{R}))$, note that

$$\mathbb{P}\left(\frac{\sum_{i \in \mathcal{L}, \mathcal{R}} 2\mu_n(A(V, \mathcal{L}, \mathcal{R}))^2}{N_n(A(V, \mathcal{L}, \mathcal{R}))} \geq \frac{m_n \delta}{3N_n(A(V, \mathcal{L}, \mathcal{R}))}\right)$$

$$\leq \mathbb{P}\left(|\mu_n(A(V, \mathcal{L}, \mathcal{R}))| \geq \sqrt{\frac{\delta m_n}{6d_n}}\right)$$

$$\leq \mathbb{P}\left(\left|\frac{1}{N_n(A(V, \mathcal{L}, \mathcal{R}))} \sum_{i=1}^{N_n(A(V, \mathcal{L}, \mathcal{R}))} \epsilon_i\right| \geq \sqrt{\frac{\delta m_n}{6d_n}} - M\right) \quad (21)$$

$$\leq 2\exp\left(-\frac{1}{2}m_n(\sqrt{\frac{\delta m_n}{6d_n}} - M)^2\right)$$

$$\leq 2\exp\left(-\frac{\delta m_n}{4}\right),$$

where the last inequality follows from $m_n \geq 8d_n$ and $\delta \geq 96M^2 d_n/m_n$. Combining Equations (18), (19), (21), we have

$$\mathbb{P}_{X, \epsilon}\left(\Delta_{\mathcal{I}}(A(V, \mathcal{L}, \mathcal{R}), k) \geq \frac{m_n \delta}{3N_n(A(V, \mathcal{L}, \mathcal{R}))}\right) \leq 5n\exp\left(-\frac{\delta m_n}{\max\{16(M+1)^2, 256\}}\right) \quad (22)$$

for any $V \subset [p], |V| = d, \mathcal{L}, \mathcal{R} \in [n+2]^{|V|}$, and $k \notin S$. Note that the set $A_d$ has cardinality

$$|A_d| = \binom{p}{d}(2(n+2))^d \leq \left(\frac{pn}{d}\right)^d$$

for large $n$. Therefore by union bound,

$$\mathbb{P}\left(\exists A \in A_d, k \notin S : \Delta_{\mathcal{I}}(A, k) \geq \frac{m_n \delta}{N_n(A)}\right) \leq 5np|A_d|\exp\left(-\frac{\delta m_n}{\max\{256, 16(M+1)^2\}}\right)$$

$$\leq 5(np)^{d+1}\exp\left(-\frac{\delta m_n}{\max\{256, 16(M+1)^2\}}\right). \quad (23)$$

Suppose that $\Delta_{\mathcal{I}}(A, k) \geq \frac{m_n \delta}{N_n(A)}$ for all $A \in \cup_{d \leq d_n} A_d$ and $k \notin S$, then for any $T \in \mathcal{T}_n(m_n, d_n)$,

$$G_0(T) \leq \sum_{t:v(t)\notin S} \frac{N_n(t)}{n} \frac{m_n \delta}{N_n(t)} \leq \delta \frac{m_n |I(t)|}{n} \leq \delta,$$

where the last inequality follows since $|I(t)| + 1$ is the total number of leaf nodes in $T$, and each leaf node contains at least $m_n$ samples. Therefore

$$\mathbb{P}_{X, \epsilon}\left(\sup_{T \in \mathcal{T}_n(m_n, d_n)} G_0(T) \geq \delta\right) \leq \sum_{d=1}^{d_n} \mathbb{P}\left(\exists A \in A_d, k \notin S : \Delta_{\mathcal{I}}(A, k) \geq \frac{m_n \delta}{N_n(A)}\right)$$

$$\leq \sum_{d=1}^{d_n} 5(np)^{d+1}\exp\left(-\frac{\delta m_n}{\max\{256, 16(M+1)^2\}}\right) \quad (24)$$

$$\leq 10(np)^{d_n+1}\exp\left(-\frac{\delta m_n}{\max\{256, 16(M+1)^2\}}\right)$$

for any $\delta > \frac{96M^2 d_n}{m_n}$. Recall that $C = 2\max\{256, 16(M+1)^2\}$. Note that $\frac{Cd_n \log(np)}{m_n} \geq \frac{96M^2 d_n}{m_n}$ for large $n$. Integrating over $\delta$, we have

$$
\begin{aligned}
\mathbb{E}_{X,\epsilon} &\left[ \sup_{T \in \mathcal{T}_n(m_n, d_n)} G_0(T) \right] \\
&\leq \frac{3d_n \log(np)}{2m_n} + \mathbb{E}_{X,\epsilon} \left[ \sup_{T \in \mathcal{T}_n(m_n, d_n)} G_0(T) \mathbb{1}(\delta \geq \frac{3d_n \log(np)}{2m_n}) \right] \\
&\leq \frac{3d_n \log(np)}{2m_n} + \int_{\frac{3d_n \log(np)}{2m_n}}^{\infty} \mathbb{P}_{X,\epsilon} \left( \sup_{T \in \mathcal{T}_n(m_n, d_n)} G_0(T) \geq \delta \right) d\delta \\
&\leq \frac{Cd_n \log(np)}{m_n}.
\end{aligned}
\tag{25}
$$

This completes the proof of the upper bound.

**Proof of the lower bound in Theorem 1**

For the lower bound, let
$$
d_n = \max\{d : 2^{d+1} m_n < n\},
\tag{26}
$$
and consider a balanced, binary decision tree $T$ constructed in the following way:

1. At each node on the first $d_n - 1$ levels of the tree, we split on feature $X_1$, at the mid-point of $X_1$'s side of the rectangle corresponding to the node.

2. At each node on the $d_n$th level, we look at the remaining $p - 1$ features, and split on the feature that maximizes the decrease in impurity.

In the following proof, we will lower bound $G_0(T)$ by the sum of impurity reduction on the $d_n$th level alone. For $t = 1, \ldots, 2^{d_n - 1}$, let

$$
R_t = \left\{ \frac{t-1}{2^{d_n-1}} \leq X_1 < \frac{t}{2^{d_n-1}} \right\}.
$$

be the hyper-rectangle corresponding to the $t$th node on the $d_n$th level. Applying Chernoff's inequality, we have
$$
\mathbb{P}\left( \left| \frac{N_n(R_t)}{n} - \frac{1}{2^{d_n-1}} \right| \geq \frac{1}{3 \cdot 2^{d_n-1}} \right) \leq 2\exp\left( -\frac{n}{27 \cdot 2^{d_n-1}} \right).
$$

Let
$$
B_1 = \left\{ \left| \frac{N_n(R_t)}{n} - \frac{1}{2^{d_n-1}} \right| \leq \frac{1}{3 \cdot 2^{d_n-1}} \text{ for all } t \right\}
$$
be the event that each node on the $d_n$th level contains at least $\frac{2}{3} \frac{n}{2^{d_n-1}}$, but no more than $\frac{4}{3} \frac{n}{2^{d_n-1}}$ samples. Then

$$
\mathbb{P}(B_1^c) \leq \sum_{t=1}^{2^{d_n-1}} \mathbb{P}\left( \left| \frac{N_n(R_t)}{n} - \frac{1}{2^{d_n-1}} \right| \geq \frac{1}{3 \cdot 2^{d_n-1}} \right) \leq 2^{d_n} \exp\left( -\frac{n}{27 \cdot 2^{d_n-1}} \right),
\tag{27}
$$

and conditional on $B_1$,

$$
\frac{8}{3} m_n \leq \frac{2}{3} \frac{n}{2^{d_n-1}} \leq N_n(R_t) \leq \frac{4}{3} \frac{n}{2^{d_n-1}} \leq \frac{32}{3} m_n.
\tag{28}
$$

We define
$$
R_t^l(k) = R_t \cap \left\{ 0 \leq X_k < \frac{1}{2} \right\}
$$
and
$$
R_t^r(k) = R_t \cap \left\{ \frac{1}{2} \leq X_k < 1 \right\}
$$

and use $R_t^l$, $R_t^r$ as shorthand when $k$ is fixed. For each $t = 0, 1, \ldots, 2^d - 1$, by Equation

$$\Delta_{\mathcal{I}}(R_t, k) \geq \Delta_{\mathcal{I}}(R_t, (k, 1/2)) = \frac{N_n(R_t^l)}{N_n(R_t)} \frac{N_n(R_t^r)}{N_n(R_t)} (\mu_n(R_t^l) - \mu_n(R_t^r))^2$$

Let

$$\eta_k = \mu_n(R_t^l) - \mu_n(R_t^r)$$

Conditional on $N_n(R_t^l)$ and $N_n(R_t^r)$, $\eta = (\eta_2, \ldots, \eta_p)$ are jointly Gaussian with zero mean. To lower bound the impurity decrease at the $t$th node on the $d_n$th level, we use a Gaussian comparison argument to obtain a lower bound for $\sup_k |\eta_k|$, which requires us to calculate the covariance matrix of $\eta$. For any $2 \leq k_1, k_2 \leq p$, let us further define

$$R_t^{ll}(k_1, k_2) = R_t \cap \left\{ 0 \leq X_{k_1} < \frac{1}{2} \right\} \cap \left\{ 0 \leq X_{k_2} < \frac{1}{2} \right\};$$

$$R_t^{lr}(k_1, k_2) = R_t \cap \left\{ 0 \leq X_{k_1} < \frac{1}{2} \right\} \cap \left\{ \frac{1}{2} \leq X_{k_2} < 1 \right\};$$

$$R_t^{rl}(k_1, k_2) = R_t \cap \left\{ \frac{1}{2} \leq X_{k_1} < 1 \right\} \cap \left\{ 0 \leq X_{k_2} < \frac{1}{2} \right\};$$

$$R_t^{rr}(k_1, k_2) = R_t \cap \left\{ \frac{1}{2} \leq X_{k_1} < 1 \right\} \cap \left\{ \frac{1}{2} \leq X_{k_2} < 1 \right\}.$$

As before, we write $R_t^{ll}, R_t^{lr}, R_t^{rl}$ and $R_t^{rr}$ as shorthand when $k_1, k_2$ are fixed. Conditional on $N_n(R_t)$, the samples falling into the hyper-rectangle $R_t$ are uniformly distributed in $R_t$. Therefore we know from Chernoff's inequality that

$$\mathbb{P}\left( \left| \frac{N_n(R_t^{ll})}{N_n(R_t)} - \frac{1}{4} \right| \geq \frac{1}{16} \right) \leq 2 \exp\left( - \frac{N_n(R_t)}{48} \right)$$

for any $k_1$ and $k_2$, and that the same results hold for $R_t^{lr}, R_t^{rl}$ and $R_t^{ll}$ as well. Let

$$B_2 = \left\{ \max_{\omega \in \{ll, lr, rl, rr\}} \left| \frac{N_n(R_t^\omega(k_1, k_2))}{N_n(R_t)} - \frac{1}{4} \right| \leq \frac{1}{16}, \text{ for all } 1 \leq t \leq 2^{d_n - 1}, 2 \leq k_1 < k_2 \leq p \right\}.$$

Then

$$\mathbb{P}(B_2^c) \leq 2^{d_n} p^2 \exp\left( - \frac{N_n(R_t)}{48} \right), \tag{29}$$

and

$$\mathbb{P}(B_1 \cap B_2) \geq 1 - 2^{d_n + 1} p^2 \exp\left( - \frac{N_n(R_t)}{48} \right) \geq 1 - 2^{d_n + 1} p^2 \exp\left( - \frac{m_n}{18} \right) \geq \frac{8}{9} \tag{30}$$

for $n$ large enough (under the condition that $m_n \geq 36 \log p + 18 \log n$). Conditional on the event $B_2$,

$$N_n(R_t^l) \geq N_n(R_t^{ll}) + N_n(R_t^{lr}) \geq \frac{3}{16} N_n(R_t) + \frac{3}{16} N_n(R_t) \geq \frac{3}{8} N_n(R_t),$$

for any $1 \leq t \leq 2^{d_n - 1}$ and $2 \leq k \leq p$, and the same holds for $N_n(R_t^r)$. Therefore,

$$\text{Var}(\eta_k) = \frac{1}{N_n(R_t^l)} + \frac{1}{N_n(R_t^r)} \geq \frac{3}{4N_n(R_t)} \tag{31}$$

$$\text{Cov}(\eta_{k_1}, \eta_{k_2}) = \frac{1}{N_n(R_t^{ll})} + \frac{1}{N_n(R_t^{rr})} - \frac{1}{N_n(R_t^{lr})} - \frac{1}{N_n(R_t^{rl})} \leq \frac{1}{4N_n(R_t)}. \tag{32}$$

Consider $\tilde{\eta}_2, \ldots, \tilde{\eta}_p$ with

$$\mathbb{E}\tilde{\eta}_k = 0, \text{Var}(\tilde{\eta}_k) = \frac{3}{4N_n(R_t)}$$

and

$$\text{Cov}(\tilde{\eta}_{k_1}, \tilde{\eta}_{k_2}) = \frac{1}{4N_n(R_t)}.$$

Then conditional on $B_1 \cap B_2$, by Sudakov-Fernique lemma, we have

$$\mathbb{E}_\epsilon[\max_k \eta_k | B_1 \cap B_2] \geq \mathbb{E} \max_k \tilde{\eta}_k \geq \sqrt{\frac{\log p}{N_n(R_t)}} \geq \sqrt{\frac{3 \log p}{32 m_n}},$$

and the lower bound

$$\min\{N_n(R_t^l), N_n(R_t^r)\} \geq \frac{3}{8} N_n(R_t) \geq m_n,$$

for any $k, t$. where the last inequality follows from Equation (28). Therefore, conditional on $B_1 \cap B_2$ the minimum leaf size is lower bounded by $m_n$. Finally

$$
\begin{aligned}
\mathbb{E}_{X,\epsilon}\left[\sup_{T \in \mathcal{T}_n(m_n)} G_0(T)\right] &\geq \mathbb{E}_{X,\epsilon}\left[\sup_{T \in \mathcal{T}_n(m_n)} G_0(T)\mathbb{1}_{B_1 \cap B_2}\right] \\
&\geq \mathbb{E}_X\left[\sum_t \frac{N_n(R_t)}{n} \mathbb{E}_\epsilon\left[\max_k \Delta_{\mathcal{I}}(R_t, k)\mathbb{1}_{B_1 \cap B_2}\right]\right] \\
&\geq \mathbb{E}_X \sum_t \frac{N_n(R_t)}{n} (\frac{3}{8})^2 (\mathbb{E}_\epsilon \max_k \eta_k^2 \mathbb{1}_{B_1 \cap B_2}) \\
&\geq \frac{9}{64} \frac{3 \log p}{32 m_n} \mathbb{P}(B_1 \cap B_2) \\
&\geq \frac{1}{80} \frac{\log p}{m_n}
\end{aligned}
\tag{33}
$$

when $n$ is large enough, and the lower bound is proved. This concludes the whole proof.

$\square$

*Proof of Proposition 1.* For simplicity, here we only present the proof for a single tree $T$. The case of multiple trees is straightforward. Recall that $t^{\text{left}}$ and $t^{\text{right}}$ are the left and right children of the node $t$. Based on (4), MDI at the node $t$ is

$$
\begin{aligned}
\frac{N_n(t)}{|\mathcal{D}^{(T)}|} \Delta_{\mathcal{I}}(t) = \frac{1}{|\mathcal{D}^{(T)}|} \sum_{i \in \mathcal{D}^{(T)}} & [y_i - \mu_n(t)]^2 \mathbb{1}(\mathbf{x}_i \in R_t) \\
& - [y_i - \mu_n(t^{\text{left}})]^2 \mathbb{1}(\mathbf{x}_i \in R_{t^{\text{left}}}) - [y_i - \mu_n(t^{\text{right}})]^2 \mathbb{1}(\mathbf{x}_i \in R_{t^{\text{right}}}).
\end{aligned}
\tag{34}
$$

Because $\mathbb{1}(\mathbf{x}_i \in R_t) = \mathbb{1}(\mathbf{x}_i \in R_{t^{\text{right}}}) + \mathbb{1}(\mathbf{x}_i \in R_{t^{\text{left}}})$, the above term becomes

$$
\begin{aligned}
\frac{1}{|\mathcal{D}^{(T)}|} \sum_{i \in \mathcal{D}^{(T)}} & \left((y_i - \mu_n(t))^2 - (y_i - \mu_n(t^{\text{left}}))^2\right) \mathbb{1}(\mathbf{x}_i \in R_{t^{\text{left}}}) \\
& + \left((y_i - \mu_n(t))^2 - (y_i - \mu_n(t^{\text{right}}))^2\right) \mathbb{1}(\mathbf{x}_i \in R_{t^{\text{right}}}) \\
= \frac{1}{|\mathcal{D}^{(T)}|} \sum_{i \in \mathcal{D}^{(T)}} & (\mu_n(t^{\text{left}}) - \mu_n(t))(2y_i - \mu_n(t) - \mu_n(t^{\text{left}}))\mathbb{1}(\mathbf{x}_i \in R_{t^{\text{left}}}) \\
& + (\mu_n(t^{\text{right}}) - \mu_n(t))(2y_i - \mu_n(t) - \mu_n(t^{\text{right}}))\mathbb{1}(\mathbf{x}_i \in R_{t^{\text{right}}}).
\end{aligned}
\tag{35}
$$

Since $\sum_{i \in \mathcal{D}^{(T)}} y_i \mathbb{1}(\mathbf{x}_i \in t^{\text{left}}) = N_n(t^{\text{left}})\mu_n(t^{\text{left}})$, we know $\sum_{i \in \mathcal{D}^{(T)}} (y_i - \mu_n(t^{\text{left}}))\mathbb{1}(\mathbf{x}_i \in R_{t^{\text{left}}}) = 0$. Similar equations hold for the right child $t^{\text{right}}$, too. Then (35) reduces to

$$
\frac{1}{|\mathcal{D}^{(T)}|} \sum_{i \in \mathcal{D}^{(T)}} (\mu_n(t^{\text{left}}) - \mu_n(t))(y_i - \mu_n(t))\mathbb{1}(\mathbf{x}_i \in R_{t^{\text{left}}})
\tag{36}
$$

$$
+ (\mu_n(t^{\text{right}}) - \mu_n(t))(y_i - \mu_n(t))\mathbb{1}(\mathbf{x}_i \in R_{t^{\text{right}}})
\tag{37}
$$

Because of the definitions of $\mu_n(t^{\text{left}})$, $\mu_n(t^{\text{right}})$, and $\mu_n(t)$, we know

$$
N_n(t^{\text{left}})\mu_n(t^{\text{left}}) + N_n(t^{\text{right}})\mu_n(t^{\text{right}}) = N_n(t)\mu_n(t).
\tag{38}
$$

That implies $\sum_{i \in \mathcal{D}^{(T)}} (\mu_n(t^{\text{left}}) - \mu_n(t)) \mathbb{1}(\mathbf{x}_i \in R_{t^{\text{left}}}) + (\mu_n(t^{\text{right}}) - \mu_n(t)) \mathbb{1}(\mathbf{x}_i \in R_{t^{\text{right}}}) = 0$.
Using this equation, (37) can be written as

$$\frac{1}{|\mathcal{D}^{(T)}|} \sum_{i \in \mathcal{D}^{(T)}} (\mu_n(t^{\text{left}}) - \mu_n(t)) y_i \mathbb{1}(\mathbf{x}_i \in \mathbb{R}_{t^{\text{left}}}) + (\mu_n(t^{\text{right}}) - \mu_n(t)) y_i \mathbb{1}(\mathbf{x}_i \in \mathbb{R}_{t^{\text{right}}}). \quad (39)$$

In summary, we have shown that:

$$\frac{N_n(t)}{|\mathcal{D}^{(T)}|} \Delta_{\mathcal{I}}(t) = \frac{1}{|\mathcal{D}^{(T)}|} \sum_{i \in \mathcal{D}^{(T)}} (\mu_n(t^{\text{left}}) - \mu_n(t)) y_i \mathbb{1}(\mathbf{x}_i \in \mathbb{R}_{t^{\text{left}}}) + (\mu_n(t^{\text{right}}) - \mu_n(t)) y_i \mathbb{1}(\mathbf{x}_i \in \mathbb{R}_{t^{\text{right}}}).$$

$$(40)$$

Since the MDI of the feature $k$ is the sum of $\frac{N_n(t)}{|\mathcal{D}^{(T)}|} \Delta_{\mathcal{I}}(t)$ across all inner nodes such that $v(t) = k$,
we have

$$\sum_{t \in I(T)} \frac{N_n(t)}{|\mathcal{D}^{(T)}|} \Delta_{\mathcal{I}}(t) \mathbb{1}(v(t) = k)$$

$$= \sum_{t \in I(T): v(t) = k} \frac{1}{|\mathcal{D}^{(T)}|} \sum_{i \in \mathcal{D}^{(T)}} (\mu_n(t^{\text{left}}) - \mu_n(t)) y_i \mathbb{1}(\mathbf{x}_i \in \mathbb{R}_{t^{\text{left}}}) + (\mu_n(t^{\text{right}}) - \mu_n(t)) y_i \mathbb{1}(\mathbf{x}_i \in \mathbb{R}_{t^{\text{right}}})$$

$$= \frac{1}{|\mathcal{D}^{(T)}|} \sum_{i \in \mathcal{D}^{(T)}} \Big[ \sum_{t \in I(T): v(t) = k} (\mu_n(t^{\text{left}}) - \mu_n(t)) \mathbb{1}(\mathbf{x}_i \in \mathbb{R}_{t^{\text{left}}}) + (\mu_n(t^{\text{right}}) - \mu_n(t)) \mathbb{1}(\mathbf{x}_i \in \mathbb{R}_{t^{\text{right}}}) \Big] y_i$$

$$= \frac{1}{|\mathcal{D}^{(T)}|} \sum_{i \in \mathcal{D}^{(T)}} f_{T,k}(\mathbf{x}_i) y_i.$$

That completes the proof. $\qquad\qquad\qquad\qquad\qquad\qquad\qquad\qquad\qquad\qquad\qquad\qquad\qquad\square$

Figure 4: The beeswarm plots for different simulation settings.

Figure 5: MDI against inverse min leaf size. This is coherent with our theoretical analysis as MDI is proportional to the inverse of minimum leaf size.

| | Deep tree (min leaf size = 1) | | | | Shallow tree(min leaf size = 100) | | | |
| | Simulated | | ChIP | | Simulated | | ChIP | |
| | C | R | C | R | C | R | C | R |
|---|---|---|---|---|---|---|---|---|
| MDI-oob | 0.762(.019) | 0.519(.018) | 0.865(.015) | 0.980(.006) | 0.748(.019) | 0.581(.019) | 0.939(.011) | 0.983(.007) |
| SHAP | 0.548(.023) | 0.325(.028) | 0.821(.023) | 0.963(.009) | 0.677(.021) | 0.462(.025) | 0.912(.015) | 0.972(.009) |
| ranger | 0.555(.034) | 0.496(.019) | 0.726(.038) | 0.974(.007) | 0.549(.034) | 0.487(.022) | 0.755(.045) | 0.985(.004) |
| MDA | 0.493(.019) | 0.507(.022) | 0.542(.025) | 0.966(.007) | 0.500(.000) | 0.577(.018) | 0.498(.006) | 0.986(.006) |
| cforest | 0.649(.029) | 0.499(.020) | 0.788(.023) | 0.929(.026) | 0.701(.033) | 0.488(.026) | 0.900(.024) | 0.979(.007) |
| MDI | 0.118(.009) | 0.092(.008) | 0.597(.023) | 0.706(.019) | 0.632(.022) | 0.397(.025) | 0.877(.020) | 0.971(.009) |

"C" stands for classification, "R" stands for regression.