[Reviews · NeurIPS 2019]

Reviewer 1



# Post-rebuttal comment Thank you for your clear and convincing answers. I am updating my score from 7 to 8. --- # Originality The main contributions are all original. While the take-home message of the study is in retrospect simple and obvious (== compute MDI importances on out-of-bag samples), the paper provides an original analysis that explains and justifies this modification of the computation of MDI importances. # Quality I admit not having checked in details the proof of Theorem 1, but this appears technically plausible given assumptions A1 and A2. Some remarks however: - I would have appreciated a controlled experiment where G0(T) can be computed exactly in order to empirically appreciate the (supposed) tightness of the bound. - Can you comment on the limitations of the results? More specifically, what if A1 and A2 are not satisfied? In real-word setups, A1 is very unlikely to hold. While most related work is properly cited, a missing reference is Section 7.2.2 of (Louppe, 2014 https://arxiv.org/abs/1407.7502) which includes recommendations similar to those formulated at the end of Section 2 of this submission. The author identifies that "feature selection bias" is caused by the misestimation of the impurity decrease. More specifically, the estimation error of the (Shannon) impurity is inversely proportional to the number of samples in the node and proportional to the cardinality of the split variable. From this, experiments conclude that early stopping mechanisms (e.g., limited depth, larger leaf sizes, etc) are a simple and effective way to reduce bias towards noisy features. I find this submission to be consistent with this hypothesis and results, which is valuable by itself. # Clarity The paper is well written and usually easy to read, provided familiarity with this line of research. # Significance This work is significant and unique. It provides i) a better theoretical understanding of MDI importances and ii) a simple but quite effective recommendation to improve their practical use.

Reviewer 2



This paper consists of two contributions. 1) it provides an interesting analysis results on the MDI feature importances of random forests, which is now one of the most widely used variable importance criteria. Their implication "deeper trees cause more bias" has a very important meaning for RF. The core idea of RF is bagging + feature bagging in each split search, which implicitly assumes that each base learner would be overfitted (fully-grown; low training errors but quite dissimilar tree structures), whereas in boosting, we usually restrict the base learners underfitted ("weak learners"). (in scikit-learn, the default options for RandomForestRegressor or ExtraTreesRegressor are max_depth=None, but max_depth=3 for GradientBoostingRegressor). 2) it also proposes the use of MDI (the mean decrease of impurity) calculated by the OOB samples (collected during bagging of RF), instead of the standard version based on the samples in the bags (bootstrapped samples). I liked this idea very much and I even think that everyone should also use this version instead of the standard ones. Each point would be very informative and nice, but the link for the proposal of 2) from 1) is a bit unclear to me. In particular, how this simple proposal can "debias" the problematic behaviors indicated in the theoretical analysis in the first part. The MDI-oob can alleviate the bias from deeper trees? Though I liked the idea of 2), it is simply the use of OOB. If the problem occurs due to the finite-sample variability between OOB samples and in-bag samples, the proposal could be somewhat natural. But the first part 1) describes the behaviors from the depths and min-leaf sizes, and how they are connected to the use of OOB is not well explained. In particular, what Proposition 1 suggests was totally unclear to me. The paper states "Without the new expression, it is not clear how one can use out-of-bag samples to get a better estimate of MDI". But it is just feeding OOB samples to each tree in RF, and just recalculate the impurities at each node, isn't it?? Whether or not the samples are from OOB, we can always calculate the impurity of any nodes, and thus, can get the MDI from any samples {(x,y)}. (though the sample-covariance viewpoint might be somewhat interesting...) The fact that the use of OOB estimate for feature selection improves the AUCs would be understandable without any understanding like Theorem 1 and fact 1. But it sounds unexplained that it is way better than MDA (which is also OOB-based estimates, isn't it?). Minor comments: - The scores and discussions for trees and forests should be more clearly distinguished. In particular, for section 4 (experiments), the number of trees in RF should be indicated. Readers can misunderstand that Figure 1 and 2 are from single trees. (I checked the code, and found n_estimators=300 though) Also, some might even think that 'bagging' can mitigate the behaviors like Figure 1 and 2. - symbols f in Proposition 1 is overlapped with f in eq (6), and confusing - line 253: Proposition 1 ==> Theorem 1...? - MDI='mean decrease impurity' is a common name? 'the mean decrease of impurity' sounds more natural? * Comments after the author response I declare that I have read the response. Thank you for making clear the questions. I understand that the link between 2) and 1) is more from a practical point of view, and we have not yet a theory for whether or not MDI-oob can "debias" MDI. I would appreciate it if you can make clear how to compute MDI-oob in the final manuscript. The response states "the bias of directly computing impurity using OOB samples could still be large for deep trees." But I'm not quite sure whether or not the proposed MDI-oob is different from directly computing ones...? If they are different, you are claiming that the proposed ones are better than "directly computing MDI-oob"..? Can we empirically compare them..? I'm a bit confused with this response, and quite happy if this point is clear in the revised manuscript.

Reviewer 3



my perspective remains unchanged after reading the author response. my original review is below. - "However, G0(T) is typically non-negligible in real data, " how could one ever know this? it requires knowing the true distribution, which we don't know for any real data - please replace tables with avg AUC with figures showing scatterplots, jittered scatterplots, or beeswarms, so we can see whether the differences are meaningful - the proof of bias is nice, and then there is a claim that the MDI-oob is debiased, but no theorem stating that, and then not even a simulation demonstrating it explicitly. please add to figures 1 & 2 the result from MDI-oob to at least provide experimental evidence for the debiased claim. - the oob idea is nice. in RF land, people have leveraged this before to get consistent estimates of uncertainty. Brieman actually suggested it in his 1984 book, in chapter 3, it is called "honest sampling". it was then seemingly independently re-discovered by http://proceedings.mlr.press/v32/denil14.pdf and http://www.jmlr.org/papers/v13/biau12a.html, and used by wager in several articles, for example, https://projecteuclid.org/euclid.aos/1547197251, and then also in https://arxiv.org/abs/1907.00325 along with another trick to decrease finite sample bias. i haven't thought about it carefully, though i imagine some of the theoretical work in those papers could form the basis of theory for your MDI-oob. probably for a future manuscript, i would recommend at least mentioning these kinds of works here. - one question i have is why couldn't one simply compute MDI on oob samples, without the new equation? i feel that needs clarification. - in the SHAP papers, they make a big deal about a few "properties" that SHAP has, that no other existing feature importance algorithms have. i'd prefer in the discussion, rather than a paragraph summarizing the results, a discussion on this topic.

[Author Response · NeurIPS 2019]

We thank the reviewers for their valuable feedback that will significantly improve our paper. We will address all the
feedback such as notations, wording, additional plots, and the missing references in the final version. For the major
comments, we organize our responses as follows.

**Reviewer 1: Limitations of the results? what if A1 and A2 are not satisfied?** While the uniform marginal
distribution assumption in A1 is relatively easy to satisfy (by transforming the features via its inverse CDF), we agree
that the independence assumption in A1 is quite strong. Correlated features are commonly encountered in practice
and difficult for any feature selection method. This is indeed a limitation of Theorem 1. We will point this out in our
revision. On the other hand, although we assume noisy features to be independent, Theorem 1 allows relevant features
to be correlated. The CHIP data included in our simulation studies shows that MDI-oob works in this setting. We would
like to deal with more general feature correlations in our future work.

**Reviewer 1: A controlled experiment where $G_0(T)$ can be computed exactly to empirically appreciate the**
**tightness of the bound.**

Our theorem states that for fixed dimension $p$, disregarding the $\log n$ terms, the bound is approximately inversely proportional to the minimum node size $m_n$. In Figure 1 of our submission, we plot the MDI importance of each feature as a function of $m_n$. If we plot the MDI importance against $1/m_n$ as in Fig. 1, then we observe that the MDI of noisy features is close to a linear function w.r.t. $1/m_n$, which verifies that the $1/m_n$ rate is tight. We plan to add this plot in our supplementary material. The constants in the proof are not tight and can be improved with a more careful analysis.

**Fig. 1. MDI for noisy features**

**Fig. 2. MDI-oob for the first sim..**

**Reviewer 2: Link between Part 2 (MDI-oob) and Part 1 (Theoretical analysis of the bias)** The link between 2)
and 1) is more from a practical point of view: 1) points out the deep tree regime where MDI has the biggest problem,
and 2) offers an empirical solution to alleviate the issue. Empirical evidence on how MDI-oob reduces the bias can be
seen in Fig. 2.

**Reviewers 2 and 3: Give theoretical/empirical evidence that MDI-oob can "debias" MDI.** Unfortunately, we do
not have theories for MDI-oob yet. Empirically, we compute the MDI-oob for the first simulation. The result is shown
in Fig. 2, which shows that MDI-oob indeed reduces the bias of MDI. We will add Fig. 2 in our future manuscript.

**Reviewer 2: Why MDA performs badly given the fact that it also uses OOB samples?** We think this could be due
to the low signal-to-noise ratio in our simulation setting. After we train the RF model on the training set, we evaluated
the model's accuracy on a test set. It turns out that the accuracy of the model is quite low. In that case, MDA measure
struggles because the accuracy difference between permuting a relevant feature and permuting a noisy feature is small.
If we increase the signal-to-noise ratio, the MDA gets better.

**Reviewers 2 and 3: Why not directly computing the impurity and the feature importance using the OOB**
**samples?** Directly computing the impurity using OOB samples may indeed lower the bias. We will add this point
in our revision. However, unless the responses of all the OOB samples falling into a node are constant, the impurity
decrease at that node is still always positive. In this case, an argument similar to the proof of Theorem 1 can show that
the bias of directly computing impurity using OOB samples could still be large for deep trees.

**Reviewer 3: "G0(T) is typically non-negligible in real data", how could one ever know this? it requires knowing**
**the true distribution, which we don't know for any real data.** Our statement requires knowing which features are
noisy in a given prediction problem. While we agree that this is generally difficult, based on our experience, there are
many applications where negative controls are measured, particularly in the biological sciences. In those problems, we
do know that such noisy features exist and thus $G_0(T)$ is often non-negligible.

**Reviewer 3: Comment on SHAP.** SHAP originates from game theory and offers a novel perspective when we analyze
the existing methods. While it is desirable to have 'consistency, missingness and local accuracy', our analysis indicates
that there are other theoretical properties that are also worth taking into account. As shown in our simulation, the
feature selection bias of SHAP increases with the depth of the tree, and we believe SHAP can also use OOB samples to
improve feature selection performance.

**All reviewers: Related work on using a validation set to compute the MDI.**

We will provide a review of related literature in our future manuscript.

[Meta-Review · NeurIPS 2019]

The paper studies theoretically the bias of the popular MDI importance measures in the presence of noisy features and proposes a very simple practical solution to reduce it. Two reviewers are very enthusiastic about the paper, even more so after reading the authors' response. One reviewer has several valid concerns about missing links between theory and practice but still recommends acceptance. I therefore recommend accepting the paper. The author are asked to take into account the reviewers comments when preparing the final version of their paper and, in particular, to address the specific request of reviewer 2 (to clarify how MDI-oob is computed).